# Spatiotemporal Dynamics of Covert Versus Overt Processing of Happy, Fearful and Sad Facial Expressions

**DOI:** 10.3390/brainsci11070942

**Published:** 2021-07-17

**Authors:** Antonio Maffei, Jennifer Goertzen, Fern Jaspers-Fayer, Killian Kleffner, Paola Sessa, Mario Liotti

**Affiliations:** 1Department of Developmental and Social Psychology, University of Padova, Via Venezia 8, 35131 Padova, Italy; antonio.maffei.phd@gmail.com (A.M.); paola.sessa@unipd.it (P.S.); 2Padova Neuroscience Center, University of Padova, Via Orus 2/B, 35129 Padova, Italy; 3Laboratory of Affective and Developmental Neuroscience, Department of Psychology, Simon Fraser University, Burnaby, BC V5A1S6, Canada; jen.barrie@gmail.com (J.G.); fern.fayer@gmail.com (F.J.-F.); kkleffner@gmail.com (K.K.)

**Keywords:** emotion, attention, ERPs, N170, EPN, LPP

## Abstract

Behavioral and electrophysiological correlates of the influence of task demands on the processing of happy, sad, and fearful expressions were investigated in a within-subjects study that compared a perceptual distraction condition with task-irrelevant faces (e.g., covert emotion task) to an emotion task-relevant categorization condition (e.g., overt emotion task). A state-of-the-art non-parametric mass univariate analysis method was used to address the limitations of previous studies. Behaviorally, participants responded faster to overtly categorized happy faces and were slower and less accurate to categorize sad and fearful faces; there were no behavioral differences in the covert task. Event-related potential (ERP) responses to the emotional expressions included the N170 (140–180 ms), which was enhanced by emotion irrespective of task, with happy and sad expressions eliciting greater amplitudes than neutral expressions. EPN (200–400 ms) amplitude was modulated by task, with greater voltages in the overt condition, and by emotion, however, there was no interaction of emotion and task. ERP activity was modulated by emotion as a function of task only at a late processing stage, which included the LPP (500–800 ms), with fearful and sad faces showing greater amplitude enhancements than happy faces. This study reveals that affective content does not necessarily require attention in the early stages of face processing, supporting recent evidence that the core and extended parts of the face processing system act in parallel, rather than serially. The role of voluntary attention starts at an intermediate stage, and fully modulates the response to emotional content in the final stage of processing.

## 1. Introduction

Human facial expressions are the main nonverbal channel for socio-emotional communication. They convey essential information regarding others’ emotional state and intentions that is critical for proper social interactions. Given their biological and social relevance, information signaled by emotional expressions should be processed rapidly in order to help regulate behavior, for example when approached by a stranger with hostile or friendly intentions.

In fact, given their importance for survival, threat-related information, such as fearful expressions, may even be processed automatically, regardless of the attentional demands imposed by the situational context [1], as a bottom-up mechanism of early surveillance to threat stimuli [2]. In contrast, when facial expressions are the focus of attention, emotional processing could be slower and more accurate due to the engagement of attentional systems, as a top-down mechanism [3].

Indeed, the interplay between attention and emotion is a key topic in the literature. On the one hand, as anticipated in the previous paragraph, stimuli of an emotional nature benefit from prioritized access to cognitive and neural resources from very early processing stages [4,5,6], more easily capture attention [7] and are resistant to attentional filtering when irrelevant [8,9].

On the other hand, the allocation of spatial [10] and top-down [3] attentional resources can modulate the processing of emotional information.

The main aim of the present investigation is to clarify the temporal dynamics of the interaction between emotion and attention by recording electroencephalography (EEG) during a within-subject design in which attentional allocation demands on the processing of different emotional expressions were manipulated by contrasting a condition of perceptual distraction with task-irrelevant facial expressions (Covert emotion task) to a condition of explicit attention to task-relevant facial expressions (Overt emotion task).

EEG offers a particularly powerful means of studying affective processing in the human brain [11,12,13] by recording neural responses to affective stimuli with temporal resolution on the order of 1–2 milliseconds (ms). Such high temporal resolution may be particularly well suited to the investigation of differences between automatic (early) vs. controlled (late) processing of face expressions.

Three ERP components have been most consistently associated with effects of emotion in faces, the N170, EPN, and LPP. The face-sensitive N170 component (140–180 ms) is recorded over occipito-temporal electrodes and is thought to reflect encoding of face structure and configuration [14,15]. The N170 has been found to be modulated by emotion. The strongest and most replicated effect is an enhancement of the N170 to fearful compared to neutral or happy faces [11,16,17,18], even in conditions of highly demanding perceptual difficulty [19,20]. A few studies have also reported N170 modulations for happy relative to neutral faces [17,18], even independent of perceptual load levels [21]. It has been suggested that other factors contribute to the N170 emotion effects including task demands during emotional processing [17,22] and the choice of reference electrodes. In fact, N170 emotion modulations are more robust in studies that employ an average reference [23,24] while, on the other end in studies that use an average mastoid reference N170 emotion modulations are often absent or replaced by positive polarity modulations over anterior, frontocentral scalp, that sometimes have an earlier latency (120–180 ms) with greater amplitudes in response to fearful compared to neutral faces [2,12,25].

A second well-established ERP marker of emotion processing is the Early Posterior Negativity (EPN, 180–350 ms), a negative-going sustained amplitude deflection distributed over occipito-temporal scalp, obtained by calculating a difference wave between conditions. Some studies have reported greater amplitude EPN effects for fearful and angry expressions compared to neutral and happy faces [13,23,26], while others have reported greater EPN modulations for pleasant relative to unpleasant pictures, with both eliciting larger responses than neutral stimuli [23,24,27]. Therefore, the EPN is thought to reflect enhanced processing of emotionally salient faces in general, with a particular sensitivity for threatening faces [23]. Similar to the N170, the EPN emotion effects appear to vary as a function of the task, suggesting a dependency on attentional requirements, and have been shown to be strongly affected by the choice of reference, with more robust EPN effects seen with an average reference [13,23]. In fact, in ERP studies employing the average mastoid reference, EPN effects were much reduced or absent, often replaced by a positive polarity modulation over frontocentral scalp, sometimes with a later latency (200–350 ms) [12,28], that has been referred to as Early Anterior Positivity (EAP) [29].

A third robust emotional ERP effect is the enhancement of the late positive potential (LPP), a sustained wave (400–800 ms) broadly distributed over posterior scalp. LPP modulation is thought to reflect increased allocation of processing/working memory resources to the motivational relevance of emotional stimuli [13,30]. As a result, LPP effects can be attenuated by top-down regulation strategies, such as suppression and reappraisal [30].

Studies have investigated the influence of task demands on emotional face processing comparing at least two tasks [22,26,31,32,33]. Three of such studies employed the same two tasks to contrast a condition of lower task demands (a gender discrimination [GD] task to address covert processing of emotional expressions) to a condition of higher task demands (an emotion discrimination [ED] task, to index overt processing of emotional expressions) [22,26,31]. The first study recorded ERPs to angry, happy, and neutral expressions using the averaged mastoid reference. The authors found that the right lateralized N170 (140–185 ms) over parieto-occipital scalp was modulated by expression only in the overt ED task, concluding that it reflected voluntary attentional modulation. In contrast, for the EPN time window (240–340 ms), they found a significant effect of expression over occipital scalp, but no evidence of modulation by task demands. Similar effects were reported for an Anterior Positivity (AP) over frontocentral scalp (160–250 ms). The authors concluded that both the EPN and the AP reflect a response to emotional salience and are not affected by voluntary attention. Later components were not analyzed. Furthermore, angry and happy expressions were combined together, preventing emotional category effects from being reported [31]. The second study also recorded ERPs to angry, happy, and neutral expressions but employed an average reference. Unlike the previous study, for the N170 they found a main effect of emotion, with angry expressions showing enhanced amplitudes relative to happy and neutral expressions, in the absence of task effects. The emotion effect continued into the EPN, with greater amplitudes to happy than neutral expressions independent of task demands. The main effect of emotion continued into the LPP, with greater voltages to angry faces than to happy and neutral expressions, with a marginal interaction with task such that emotion effects were enhanced in the overt task [26]. Finally, a third study recorded ERPs to fearful, happy, and neutral facial expressions and also employed the average reference [22]. As per a previous study [26], they found that the N170 and EPN were modulated by emotion. N170 amplitudes were more negative for fearful than neutral expressions, independent of the task demand, with no detectable effect for happy expressions. In contrast, the EPN was affected by task demands, with more negative voltages for the overt than the covert task, independent of emotion [22]. Bear in mind though, that, as in the first study [31], late effects (spanning the LPP component) were not analyzed. In conclusion, results of these three studies varying task demands with the GD and ED task, as well with other studies comparing at least two tasks [22,26,31,32,33], are inconsistent in terms of which ERP components show effects of emotion or task demands i.e., whether and when top/down attentional modulation takes place; and which emotions are modulated or prioritized and at what stage.

An additional limitation of the three studies [22,26,31] is the contrast between the GD task (covert) and the ED task (overt). In the GD task participants still have to focus their attention on the face. Even without using stimuli displaying extreme masculine/feminine features, emotional expressivity and gender might be confounded due to implicit stereotyping mechanisms [34,35], therefore it is not clear whether GD relies on a different set of features than the emotional expression [36]. Furthermore, according to the distributed model of face perception [37], processing of face identity (invariant features) and emotional expression (variant features) both rely on the activation of partly overlapping regions, comprised in the extended system [37]. Lastly, among the various emotional face databases available, stimuli are typically selected on the basis of expression and not gender discriminability, therefore this latter component may not be as optimally controlled.

A better choice to study the effect of top-down attentional deployment to facial expressions could be to contrast a perceptual distraction task in which expressions are task-irrelevant (Covert condition) to an “Attention to Emotion” task where facial expressions are task-relevant (Overt Condition). These two conditions may be better matched in terms of general task demands and/or the number of stimulus and response choices than the GD and ED tasks [22,26,31]. In fact the GD task involves a two-choice selection (male/female), while the ED task requires a three- (or more) choice selection. As a result, in the present study, we employed as a Covert task a perceptual distraction task in which emotional expressions were shown with superimposed colored squares and the instructions were to respond to the color of the square ignoring the surrounding faces. In the Overt task, the exact same stimuli were shown while the instructions were reversed, i.e., to categorize the emotional expression while ignoring the color of the square.

Finally, in the present study we employed a state-of-the-art non-parametric permutation-based statistical framework [38,39] in order to possibly overcome some of the shortcomings inherent to the statistical approaches used in previous studies to model event-related potentials [40]. The mass-univariate design applied to all scalp electrodes and all time points had the potential to fully characterize the spatiotemporal aspects of the ERP response to the emotional expressions while avoiding the constraints of an arbitrary selection of specific electrode sites or regions of interest, ultimately returning significant clusters of sensors where scalp voltage significantly differed as a function of emotion, covert-overt task demands, and their interaction.

## 2. Methods

### 2.1. Participants

Sixteen participants were enrolled for the study (*M*_age_ = 19.7 ± 1.56 years; 50% female). All volunteers gave their written informed consent before participating in the study and received course credit for their participation. Participants completed a medical screening and demographic questionnaire, and all participants reported no history of neurological or psychiatric disorders, nor drug or alcohol abuse, and no learning disabilities. Participants had normal or corrected-to-normal visual acuity, normal color vision, and they were all right-handed. The study received approval from the Simon Fraser University ethics committee.

### 2.2. Experimental Task

The experiment used colored photographs of faces taken from the Karolinska stimuli set [41] (13 males, 15 females; 4 emotions: fear, sad, happy, and neutral). The faces were set on a black background and altered using Photoshop (version 10.0.1, Adobe Inc., San Jose, CA, USA) to obscure the hairline and create identical facial contours. Then each face had a small colored square (red, blue, green, or yellow) superimposed on the nose. Faces were presented for 200 ms, followed by a fixation-cross presented for a randomly jittered inter-stimulus interval (ISI) of 1700 to 2300 ms (see Figure 1). This stimulus duration was chosen to optimally minimize eye movements, that have a latency of at least 180 ms [22]. The experiment was coded using Presentation software (Neurobehavioral Systems Inc., Berkeley, CA, USA). The study included two tasks with identical stimuli but different instructions. In the Covert emotion task, participants were asked to attend to the central square irrespective of the surrounding faces, and to choose the color as quickly and accurately as possible (blue, red, green or yellow) by pressing one of four corresponding buttons on a gamepad controller with the index or middle finger of the left or right hand (Logitech, Romanel-sur-Monges, Switzerland). In the Overt emotion task, participants were asked to categorize the expression (happy, fearful, sad or neutral) conveyed by the face, irrespective of the color of the central square, by pressing as quickly and accurately as possible one of the four corresponding buttons on the gamepad controller. To help minimizing eye movements, participants were also instructed to keep their eyes on a central fixation throughout the experiment. Stimuli were displayed in a pseudo-randomized order, constrained so that no more than three stimuli with the same emotion, color, or gender were presented in a row. Participants completed a practice block and were required to reach 80% accuracy before advancing to the actual experiment. Each task (Covert vs. Overt) included 400 stimuli divided into four five-minute blocks separated by short resting periods. To control for potential confounds, button assignments relative to colors (Covert task) or emotions (Overt task) were counterbalanced between participants. The presentation order of the Overt and Covert tasks was also counterbalanced across participants. For each subject, reaction times (RTs) were recorded from stimulus onset and averaged for each combination of condition and emotion. RTs shorter than 200 ms likely corresponded to guesses or trials with eye movements and were discarded from analysis. RTs longer than 2000 ms were discarded as well due to button-press errors.

### 2.3. EEG Data Acquisition

Data were collected using high-density EEG during the performance of the Covert/Overt face emotion task. Participants sat in a sound-attenuated booth with standardized ambient lighting facing a CRT monitor positioned 60 cm away from the participants’ eyes. The ActiveTwo BioSemi electrode system (BioSemi; Amsterdam, The Netherlands) was used to record continuous EEG from 136 Ag/AgCl electrodes, 130 embedded in an elastic cap and positioned in a modified 10–20 equiradial layout relative to the vertex, including two sensors replacing the “ground” electrodes, i.e., the common mode sense (CMS) active electrode, and the driven right leg (DRL; (BioSemi; Amsterdam, The Netherlands). Six additional external electrodes were applied: two at each lateral canthus (HEOG; for horizontal eye movements), two below each eye (VEOG; for vertical eye movements and blinks), and two over each mastoid bone. DC offset was kept below +/−25 KΩ. The continuous signal was acquired with an open pass-band from DC to 150 Hz and digitized at 512 Hz. The amplifier gain was fixed for each active electrode channel at 32×.

### 2.4. EEG Preprocessing

Continuous data were high-pass filtered at 0.01 Hz, re-referenced to the average reference, and segmented in epochs starting from −200 ms to 800 ms with respect to stimulus onset. Bad channels were detected using the automated algorithm provided by the TBT plugin available in EEGLAB. The TBT algorithm detects, on a trial-by-trial basis, channels with an amplitude exceeding ±250 μV and marks them as bad. Then, channels marked as bad on more than 30% of all trials and epochs with more than 10 bad channels are rejected. Additionally, epochs with a peak-to-peak amplitude exceeding ± 100 μV in any channel were identified using a moving window procedure (window size = 200 ms, step size = 20 ms), to discard from further analysis epochs contaminated by eye-blinks and movement-related artifacts. Finally, epochs with step-like artifacts of an amplitude exceeding 50 μV were identified using a moving window procedure (window size = 400 ms, step size = 20 ms) and discarded. The average percentage of epochs retained per condition was 68%. The artifact-free epochs were low-pass filtered at 30 Hz and separate ERP averages were obtained for all combinations of Condition (Covert vs. Overt) and Emotion (happy, sad, fearful and neutral), time-locked to stimulus onset, and baseline corrected using a 200 ms baseline. The preprocessing was performed in MATLAB (v2019b) using functions from the EEGLab (v.2020.0 [42]) and ERPLab (v8.0 [43]) toolboxes.

### 2.5. Statistical Analysis

#### 2.5.1. Behavioral Data

Mean accuracy and RT for each subject were analyzed via Repeated Measures ANOVAs, with Condition (Covert vs. Overt) and Emotion (Happy, Sad, Fear, Neutral) as within-subject factors. Post-hoc paired-samples t-tests were run to test the main effects or interactions with a Bonferroni correction to control for family-wise error (alpha level set at *p* < 0.05).

#### 2.5.2. ERP Data

Statistical inference for ERP data was implemented in a mass-univariate framework [44]. This framework consists in performing a statistical test for every point in the electrode by time plane by repeatedly permuting through conditions. Repeating this process for a sufficient number of permutations estimates the empirical null-distribution of the test statistic, which can be used for inference [44]. Given the design of the present experiment, we used the factorial mass-univariate testing (FMUT) approach [39], which represents an extension of the classical mass-univariate framework able to encompass factorial ANOVA designs.

Based on previous literature, we defined three a priori time-windows of interest: Early, 140–180 ms, spanning the N170; Intermediate, 200–400 ms, including the EPN; and Late, 500–800 ms, encompassing the LPP. Then, for each time window we fit a mass-univariate ANOVA with 5000 permutations that included the predictors Emotion (4 levels: Fear, Happiness, Sadness, and Neutral), Condition (2 levels: Covert and Overt), and their interaction. The multiple comparisons problem was handled using the cluster-based approach [38]. When the FMUT revealed a significant effect, it was further explored using post-hoc mass-univariate *t*-tests, run with 5000 permutations and corrected for multiple comparisons using the cluster-based approach [44].

## 3. Results

### 3.1. Behavioral Data

#### 3.1.1. Accuracy

Participants were highly accurate in both conditions (at least 84%). A repeated-measures ANOVA revealed a significant main effect of Emotion (*F*_3,39_=18.24, *p* < 0.001, partial *η*^2^ = 0.58). This main effect was qualified by the significant Condition × Emotion interaction (*F*_3,39_ = 14.96, *p* < 0.001, partial *η*^2^ = 0.53). Post-hoc paired-samples *t*-tests (α = 0.05/6 = 0.008) revealed that participants were less accurate when responding to explicitly presented fearful and sad faces relative to explicitly presented neutral faces (*t*_15_ = −3.96, *p* = 0.001, Cohen’s *d* = −1.13; *t*_15_ = −4.17, *p* = 0.001, Cohen’s *d* = −1.32). No other contrasts were significant.

#### 3.1.2. Reaction Time

The repeated-measures ANOVA returned significant main effects for Condition (*F*_1,13_ = 27.99, *p* < 0.001, partial *η*^2^ = 0.68) and Emotion (*F*_3,39_ = 31.64, *p* < 0.001, partial *η*^2^ = 0.71), both qualified by a significant Condition × Emotion interaction (*F*_3,39_ = 23.89, *p* < 0.001, partial *η*^2^ = 0.65). RTs to Overt faces were on average 127 ms slower than those to Covert Faces (851.77 vs. 724.65 ms). Post-hoc paired-samples *t*-tests were performed on the significant Condition × Emotion interaction, with the α level corrected for 6 comparisons (α= 0.05/6 = 0.008). In the Overt condition, participants responded significantly slower to fearful faces (*t*_15_ = 4.93, *p* < 0.001, Cohen’s *d* = 0.85) and sad faces (*t*_15_ = 3.67, *p* = 0.002, Cohen’s *d* = 0.55) relative to neutral faces, and responded faster to happy faces than neutral expressions (*t*_15_ = −4.36, *p* = 0.001, Cohen’s *d* = 0.63). In contrast, in the Covert condition, there were no significant differences between emotions. Mean RTs for the combination of Condition and Emotion are summarized in Table 1.

### 3.2. ERP Results

#### 3.2.1. Early Time Window (140–180 ms)

In this epoch, encompassing the N170, the FMUT returned a significant cluster for the Emotion main effect (F cluster mass = 4514.12, *p* < 0.0001, partial *η*^2^ = 0.225), while no significant clusters were detected for the Condition main effect, nor the Condition × Emotion interaction. Post-hoc pairwise mass-univariate *t*-tests (shown in Table 2 Top and Appendix A) revealed that, independent of Condition (Covert or Overt), significantly greater ERP amplitudes were elicited by Happy compared to Neutral faces (Figure 2, top left). The significant N170-like negative cluster included scalp sites over right lateral occipital and inferior parietal scalp. The significant positive cluster, similar in timing but with a positive polarity (here called pP1), had a scalp distribution over left ventrolateral frontal and temporal scalp. Furthermore, the contrasts between Happy and Sad and Happy and Fearful faces both revealed significant negative clusters (greater N170 for Happy Faces), with a more dorsal bilateral distribution over parietal scalp (Figure 2, bottom).

Finally, Sad expressions elicited significantly greater ERP voltages relative to Neutral faces (Figure 2, top right). A significant N170-like negative cluster included electrode sites over bilateral occipital and inferior parietal scalp. A significant positive cluster, similar in timing but with a positive polarity, had a scalp distribution over anterior midline and lateral frontal scalp.


Conventional grandaverage waveforms overlaid for each Emotion for the Covert and Overt condition are shown in Appendix A (displaying a representative array of 25 scalp sites) and Figure 5 (showing the N170 effects at the posterior and anterior electrode sites of larger amplitude).

#### 3.2.2. Intermediate Time Window (200–400 ms)

In this window, spanning the EPN, the FMUT revealed both a significant cluster for the main effect of Condition (Figure 3 top; F cluster mass = 152067.2, *p* < 0.0001, partial *η*^2^ = 0.47), and a significant cluster for the main effect of Emotion (Figure 3 bottom; F cluster mass = 8457.2, *p* < 0.0001, partial *η*^2^ = 0.19) but no significant clusters were detected for the Condition × Emotion interaction.

Post-hoc pairwise mass-univariate *t*-tests on the Condition main effect (shown in Table 2 center and Appendix A) revealed that, independent of Emotion, significantly greater ERP amplitudes were elicited by Overt compared to Covert faces (Table 2). The significant negative cluster, similar to the EPN, extended to a broad region over posterior scalp, including bilateral occipital and parietal sites. The significant positive cluster, similar to the early anterior positivity (EAP), had a broad anterior scalp distribution including frontal and central sites, with a maximum over dorsal frontal scalp, both midline and left lateral.

Post-hoc pairwise mass-univariate *t*-tests on the Emotion main effect did not reveal significant differences among expressions. In order to achieve more power and to better characterize the Emotion main effect, we performed a mass-univariate t-test between the average of the three emotions and the neutral expression, (i.e., Emotion vs. non-Emotion) which did demonstrate a significant negative posterior cluster, similar to the EPN, over inferior bilateral occipital and ventral temporal scalp.

Appendix A shows the EPN effects employing conventional grandaverage waveforms for each emotion and condition at the posterior and anterior electrode sites of larger amplitude.

#### 3.2.3. Late Time Window (500–800 ms)

In this time window, encompassing the LPP, the FMUT revealed a large significant cluster for the main effect of Condition (Figure 4B; F cluster mass = 182434.4, *p* < 0.0001, partial *η*^2^ = 0.475), a significant cluster for the main effect of Emotion (Figure 4A; F cluster mass = 8783.9, *p* = 0.01, partial *η*^2^_p_ = 0.204), and, unlike the preceding time windows, a significant cluster for the Emotion x Condition interaction (Figure, 4C; F cluster mass = 5897.3, *p* = 0.04, partial *η*^2^ = 0.20). This pattern of results was further explored by post-hoc mass-univariate *t*-tests on the Condition × Emotion interaction (see Table 2 Bottom and Appendix A).

First and unsurprisingly, significantly larger amplitudes were evident for faces in the Overt relative to the Covert task (Table 2, bottom). Of particular interest, there were significant contrasts between emotions in the Overt condition, whereas there were no differences between emotions in the Covert task. For the Overt task, Happy expressions elicited significantly reduced voltages compared to Fear, Sad and Neutral, with a positive cluster (corresponding to the LPP) over centroparietal scalp that was maximal over right parietal scalp (please note that the direction of the contrast was inverted for Neutral faces, yielding a negative cluster). The effect was strongest for the Fear vs. Happy comparison (Figure 4A), where a negative cluster was also significant over anterior lateral scalp.

Appendix A shows the LPP effects employing conventional grandaverage waveforms for each emotion and condition at two posterior electrode sites of larger amplitude).

## 4. Discussion

Behavioral and electrophysiological correlates of covert and overt processing of happy, fearful and sad facial expressions were investigated by comparing a perceptual distraction condition with task-irrelevant faces to an emotion discrimination condition and applying a non-parametric mass univariate approach to address some of the limitations of previous ERP studies.

Behaviorally, participants responded faster to overt happy faces and were slower and less accurate to sad and fearful faces. In contrast, there were no differences between expressions in the covert condition. In the ERP response to the emotional expressions, a N170-like enhancement by emotion was found, irrespective of the covert-vs-overt task demands, with no indication of a voluntary attention effect at this early stage. Furthermore, the N170-like emotion effect was explained by greater amplitudes for happy expressions than neutral or fearful, as well as more negative voltages for sad faces than neutral. At an intermediate latency, the EPN and its positive counterpart the EAP were modulated by task demands, with greater amplitudes when faces were overtly attended. Furthermore, the EPN was modulated by expression, independent of emotion type or valence, and irrespective of whether the faces were attended voluntarily or not. Only at a late processing stage, the LPP, were amplitudes modulated by emotion as a function of task demands, with greater amplitude enhancements for overtly processed fearful and sad faces than happy faces.

### 4.1. Behavioural Results

Overall, participants were slower in the Overt Emotion Discrimination Task than in the Covert Emotion task, likely reflecting deeper perceptual processing in the former. Faster reaction times for happy relative to all other expressions in the Overt Emotion task is similar to the effects reported for the ED task [22,26,31], and consistent with the body of behavioral research using explicit categorization tasks that has consistently shown a recognition advantage for happy faces compared to all the basic expressions, both in accuracy and efficiency and across response modalities (manual, verbal, and saccadic) [45]. Similarly, the lack of behavioral differences among emotions in the Covert Emotion task has been previously found for covert emotional processing using the GD task [22,26,31].

### 4.2. ERP Results

#### 4.2.1. Early Time Window (140–180 ms)

At the early stage of the face-sensitive N170, amplitude was significantly modulated by emotion, as consistently reported in prior research [11,22,23,24]. In contrast, voltage in this early epoch did not vary as a function of covert-overt task demands and, importantly, the emotion enhancement took place irrespective of covert-vs-overt processing, with no indication of top-down voluntary attentional modulation.

The high-density electrode array combined with the mass-univariate approach that is not constrained by the choice of a specific electrode or region of interest, allowed characterization of the early ERP modulation at the whole scalp level, thus including both a negative modulation with a temporo-occipital scalp distribution resembling the N170 topography as is more typically reported [11,22,23,24], and a positive modulation with an anterior frontocentral distribution, as reported in some ERP studies of emotional processing employing the average mastoid reference. In such studies, the early latency of such anterior positivity (120–180 ms) has been interpreted to reflect the rapid evaluation of the emotional significance of the facial expression by prefrontal regions [2,12,25,46]. This anterior emotional effect appears to be a modulation of the pP1, an early positive polarity wave over prefrontal scalp co-occurring with the occipital N1 observed in visual ERP studies [47]. It is important to note that the scalp topography of the early anterior positivity in this study and others does not correspond to that of a similar latency positive polarity wave, the vertex-positive scalp potential (VPP) elicited by faces and objects [48].

The pattern of results in this study differs from a previous study contrasting the GD and ED task [31], which employed a mastoid reference but only reported a right N170 modulation by expression in the Overt ED task, ultimately concluding that it reflected voluntary attentional control [31]. More in line with our results are those of two other studies comparing the GD to the ED task that focused on the N170 at parieto-occipital sites. They reported a main effect of emotion as well, not affected by task demands [22,26]. Combining the available evidence, it is reasonable to conclude that the early stage of the N170 emotion enhancement and its anterior positive counterpart reflect a more automatic and bottom-up influence of emotional face content, and appear insensitive to the deployment of top-down attentional resources required by explicit emotion discrimination. This is in line with the conclusions of other studies in which the N170 emotion effect was interpreted as signaling the extent of acquired emotional information, independent of the level of perceptual load difficulty [49,50].

However, our results differ from previous studies in terms of which emotion was modulated in the early stages independent of task demands [22,26]. Among the studies varying task demands with the GD and ED task, one study found that angry faces were modulated at the N170 stage relative to happy and neutral faces [26]. A second study reported that fearful faces yielded a N170 enhancement relative to neutral expressions, with no detectable effect for happy expressions [22]. The present study found that the emotion categories enhanced at an early stage, independent of attention allocation, were happy and sad expressions.

Our study found strong evidence of an early ERP modulation to happy expressions relative to neutral and other emotions, independent of task demands and voluntary deployment of attentional resources. While the majority of ERP face studies have reported fear-related N170 effects [11,16,17,18,19,20,21], only a few ERP studies have found similar happy-neutral increases [11,17,18,21,45,51]. To our knowledge, only one recent study reported N170 happy modulation that was unaffected by perceptual load level [21].

The ERP and behavioral results of the present study could provide further support for an enhanced perceptual encoding mechanism to prioritize the processing of happy faces that may explain the behavioral advantage of happy faces in recognition tasks [45]. We can speculate that such happy face advantage may be relevant to the adaptive functions fulfilled by happy faces both in social interaction [52] and at an intrapersonal level [53]. 

Furthermore, our study found new evidence of an early N170 increase to sad faces relative to neutral faces, independent of the top-down attentional requirements. Very few ERP face studies have included sad faces, and have reported significant early effects of sad faces, along with other expressions [11,54]. The available literature concerning sad expressions pertains mostly to their use for assessing negative cognitive bias in clinical and trait depression, where depressed participants have been found to have greater N170 (or anterior positivity) responses to sad faces relative to neutral faces, while non-depressed groups show N170 increases to happy relative to neutral faces [55,56]. In the former study, N170 amplitudes to sad faces correlated with depression severity [55].

In our study, from a medical questionnaire, participants reported no present or past psychiatric conditions, including depression or anxiety. Our study provides evidence for an early prioritization of sad faces, although unlike happy faces, sad faces were categorized more slowly and less accurately than neutral faces in the overt task. We could speculate that early processing of sad faces relates to the important function of sad expressions as cues in social communication, acting as self-evaluative social feedback or facilitators in social interactions [57].

An unexpected finding in our study was the lack of an early N170 modulation for fear expressions, particularly in light of a corresponding effect for equally distressing sad faces. As reported above, fear-related N170 increases are the most robust and replicated effect, independent of task-requirements [11,16,17,18,19,20,21,22]. We do not believe this may be due to a lack of power due to sample size, since reliable effects were present for other emotion categories. Another explanation could be that fearful and sad faces were presented within the same tasks, and studies have shown that they are hard to discriminate from each other [58]. However, in our study, both sad and fearful faces were categorized slower and less accurately than neutral faces, although speed and accuracy were similar among the two (and accuracy rate was above 84%). If sad and fearful faces were grouped together in the same broad category, we would have expected less accuracy, and a similar lack of early ERP modulation for both fearful and sad images. Another possibility is that in spite of the relatively high response accuracy, the specific fear images selected for the present study may have been harder to recognize or categorize relative to other expressions and relative to fear expressions in other studies. It is worth noting, furthermore, that both sad and fearful images did modulate the late LPP wave (see below), and more so for the overt attention task, providing evidence that top-down attentional deployment was indeed necessary to discriminate both sad and fearful faces.

A plausible explanation of the N170 emotion effects in the present study and the apparently conflicting pattern of emotion categories being prioritized in the previous studies could be drawn by the evidence that the N170 has been found to be sensitive to perceived arousal of facial expressions rather than categories of emotional expressions [59]. In light of this evidence, it is possible that the apparent contrast between previous findings and our results regarding the modulation of the N170 and the anterior positivity as a function of the categories of emotional expressions may be attributable to heterogeneity in the perceived arousal for different categories of emotional expressions between the different studies. Interestingly, the perceived arousal account fits well with the ERP studies of emotional processing of sad faces in individuals with trait or clinical depression [55,56], where the augmented N170 to sad faces in depressed individuals can be explained by the increased personal salience of sad, symptom-related stimuli, as well as similar ERP studies demonstrating hypervigilance to threat stimuli in anxious individuals [7,60]. This interpretation also fits well with N170 emotion accounts in term of accrued emotional information, where neutral faces paired with negative biographic information or negative “outgroup” faces yielded N170 increases [49,50].

A final point worth discussing concerns the different scalp topographies of the early ERP effects to happy and sad faces. While the N170-like negative cluster was similar in spatial distribution over temporo-occipital scalp, the positive cluster had distinctly different scalp topographies over anterior scalp. In response to happy faces, the early anterior positivity response significantly differed from that to neutral faces over left inferior lateral prefrontal and temporal scalp. This laterality appears consistent with a large body of EEG research across different age groups showing that people who are generally happy tend to have greater neural activity (alpha desynchronization) over the left frontal scalp, while people that tend to be sad or are clinically depressed tend to have greater activity over the right frontal scalp areas [61]. The present results may suggest that the presentation of smiling faces in healthy individuals may very rapidly prime or activate left lateralized corticolimbic networks involved in the generation and regulation of positive affect, and motivate the individual toward a prosocial, approach action tendency.

In contrast, in response to sad faces, the early anterior positivity response significantly differed from that of neutral faces over anterior frontomedial and bilateral frontolateral scalp. The greater involvement of medial prefrontal cortex in the detection or appraisal of sad faces may relate the sensitivity of this subregion to self-relevant information and evaluative social feedback [62], to social rejection [63], and the subjective experience of sadness [64].

#### 4.2.2. Intermediate Time Window (200–400 ms)

Unlike the early time window, for the intermediate epoch containing the EPN, there was a main effect of task demands, with greater voltage in the Overt than Covert emotion task, as well as a main effect of Emotion, but no evidence that the EPN emotion enhancement was affected by covert-vs-overt processing, i.e., no indication of top-down voluntary attentional modulation.

The pattern of results in the present study differs from the one reported by two of the previous studies employing the GD and ED tasks [26,31], that reported an emotion modulation, but no effect of task demands on this component, concluding that the EPN is not affected by voluntary attention. In contrast, our EPN results are in line with those of the third study [22] which reported both an EPN enhancement as a function of emotion, and an EPN-like modulation as a function of task demands, with more negative voltages over temporo-occipital sites for the Overt than the Covert task, but this effect was independent of Emotion.

Concluding from the latter study and ours, it appears that after the N170, in the EPN time window, cognitively mediated top-down attentional control modulates neural activity, presumably reflecting the depth of processing of all face stimuli in the explicit emotion discrimination task, independent of emotion. The significant EPN emotion effect in the present study did not survive post-hoc testing based on emotion categories, and was only apparent when all emotional expressions were contrasted to the neutral faces. This may be explained by an increase in power by combining all emotions together. The significant negative cluster obtained from the mass univariate testing extended over bilateral temporo-occipital scalp, while there was no significant positive cluster counterpart for this effect. The EPN emotion modulation is similar to that reported in one of the three studies contrasting the GD to the ED task, which similarly collapsed emotional expressions in the ED task [31]. In another such study the EPN modulation was present for happy expressions relative to neutral faces [26], while in a third study EPN amplitudes were more negative for fearful than neutral expressions, with no detectable effect for happy expressions [22]. The presence of EPN modulations both for negative and positive emotions relative to neutral stimuli was reported also in a number of other studies [23,24,27]. This is consistent with the conclusion that the EPN may reflect enhanced processing of emotional salience of stimuli in general, likely related to meaning in terms of personal salience rather then to a perceptual encoding mechanism [13].

Concerning the EPN modulation by task demands in the present study, the mass univariate analysis returned a significant negative cluster, extended to a broad region over posterior scalp, including not only the traditional EPN occipito-temporal scalp sites but also parietal sites. In addition, there was a large positive cluster with a broad anterior scalp distribution including frontal and central sites maximal over dorsal frontal areas, both midline and left lateral. This positivity appears similar in timing and scalp topography to the early anterior positivity (EAP), a positive voltage modulation to emotional vs. neutral stimuli [29,60] in studies employing the average mastoid reference, that appears to be slightly delayed with respect to the EPN (see also [12,28]).

#### 4.2.3. Late Time Window (500–800 ms)

The main effect of task demands continued in the late time epoch, with greater amplitude LPP to overtly processed than covertly processed facial expressions. This is not surprising given the fact that the LPP is thought to reflect increased allocation of processing resources to motivationally relevant stimuli [13,30], and that the LPP emotion effects are attenuated by top-down regulation strategies, such as suppression and reappraisal [30]. More interesting was the finding that at this late stage the effect of task demands varied as a function of emotion. While no LPP voltage differences were present among the emotions in the covert emotion task, for the Overt ED task, Fear and Sad expressions elicited greater LPP amplitudes than happy faces. The results of the study suggest that the allocation of processing resources and top-down attentional control to emotional expressions in the overt emotion task operates at the late stage of the LPP. In the previous ERP studies of emotional face processing examining tasks demands with the GD and ED task, two of the relevant earlier studies did not report late ERP effects [22,31]. The third one reported an emotion effect, with greater LPP voltages for angry faces than happy and neutral faces, and only “a marginally significant” interaction, with greater Emotion effects in the Overt task [26].

Particularly interesting were the findings that fearful and sad faces produced ERP emotion enhancements at this late stage when top-down attentional resources were applied in the overt emotion task. This stands in contrast to the early modulation observed in response to happy and sad faces, independent of task demands, suggesting a prioritization of these emotional expressions. The early bias toward fearful faces may become more apparent in anxious individuals, reflecting the well demonstrated hypervigilance to threat in anxiety [2,60]. Two previous relevant studies contrasting the GD and ED task did not include fearful faces [26,31]. The third relevant study did include fearful faces, but they did not report on LPP effects, while they found fear-related modulations for the N170 and the EPN as many other studies in the literature. In contrast to our study, the authors reported no happy face effects at any latency [22]. It is not clear why the latter study and the present one yielded such discrepant findings. The best account, as mentioned earlier, is that perceived arousal of a given emotion category may have differed across studies. Some other methodological differences may have contributed to such discrepancy. The covert task employed in that study (the GD task) was different, and perhaps not equally well matched. The combination of expressions was different (fear, happiness and neutral), and did not include sad faces. Furthermore, they employed the Nimstim dataset rather than the Karoliniska dataset [22] and the expressions were therefore not entirely comparable, contributing to differences in perceived arousal and personal saliency.

## 5. Conclusions

In regard to the interplay between attention and emotion, this study provides some interesting findings that add to the ongoing debate regarding the complex dynamics of face processing [65,66]. This study reveals that affective content does not necessarily require attention in the early stages of face processing (<200 ms). This supports recent evidence that the core and the extended part of the face-processing system act in parallel, rather than serially, and continuously exchange information [65,66]. The role of voluntary attention starts at an intermediate stage and fully modulates the response to emotional content in the final stage of processing. Indeed, it is only after the face has been fully encoded that the cognitively mediated attentional control over its affective cues takes place. This leads to the selection and the enhancement of emotional information in general, and aversive content specifically [67]. The present result is also in accordance with a very recent study that addressed the cortical time course of the interaction between attention and emotion during face processing [68]. This convergence further stresses how important is it for characterizing the dynamics of affective processing to not restrict the investigation to a single time-windows/components-of-interest. A comprehensive analysis of the full time-course of the neural activity over the entire scalp surface captures the spatiotemporal dynamics of the attention/emotion interaction. Incidentally, a very recent ERP re-analysis of an emotion face-study contrasting the GD and ED task with a similar mass univariate approach showed similar effects as our study, although the time-window examined was restricted to the first 350 ms and EEG was recorded from 64 rather than 128 channels [69].

### Limitations and Future Directions

The inferences in terms of underlying brain regions derived from the scalp topographies of the significant spatiotemporal clusters identified by the mass univariate approach should be interpreted with caution, since no modeling of brain electrical sources was carried out.

Our choice of face expressions was slightly biased in terms of gender representation (15 females, 13 males), because we discarded some male expressions we did not find as convincing. Power considerations prevented us from assessing the influence of the gender of the portrayer.

The present study could have benefited by a larger sample size, which may have allowed further consideration of the influence of important variables such as gender of the portrayed face, which has been shown to bias emotion categorization [34,35], as well as gender of the observer. Sex differences in emotional processing have been reported both at the behavioral and the level of brain structure and function [70,71]. A larger sample size of healthy participants could have also allowed to take into consideration the impact of individual differences in psychological traits or measures relevant to emotional processing, such as trait and state depression, trait and state anxiety [60], impulsivity, personality traits such as extroversion vs. introversion, empathy, alexithymia, etc.

Finally, it should be noted that the mass univariate approach, while allowing a more comprehensive analysis of the full time-course of the neural activity over the entire scalp surface in an unbiased way, does present some trade-offs relative to the traditional ERP approach based on time windows and predefined scalp sites. While it is less sensitive to small effect sizes, it is more able to detect activity from distributed than focused neural sources, and it is relatively insensitive to latency information.

These limitations notwithstanding, this is to our knowledge one of first studies to explore within-subjects ERP correlates of processing of face expressions as a function of attentional deployment to emotional faces, contrasting a “perceptual distraction” to an “attention to emotion” condition, and to report ERP evidence that affective content does not necessarily require attention in the early stages of face processing and that the role of voluntary attention would start in an intermediate stage and then fully modulate the response to emotional content only in the final stage of processing.

## Figures and Tables

**Figure 1 brainsci-11-00942-f001:**
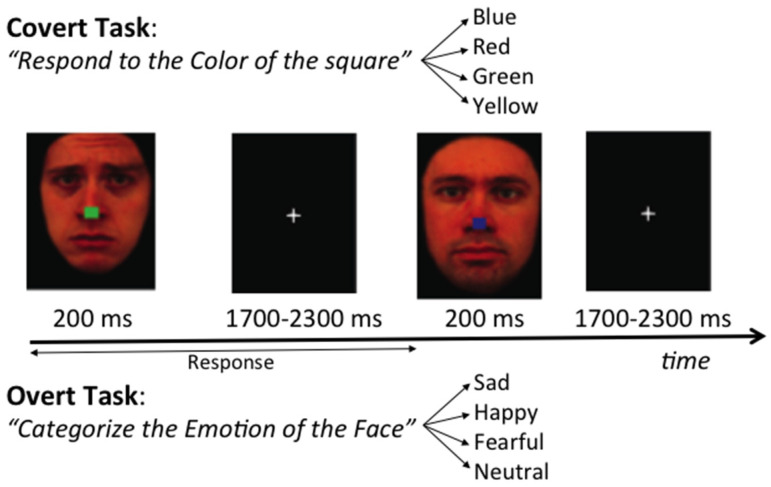
Trial sequence in the Covert and Overt Emotion Tasks.

**Figure 2 brainsci-11-00942-f002:**
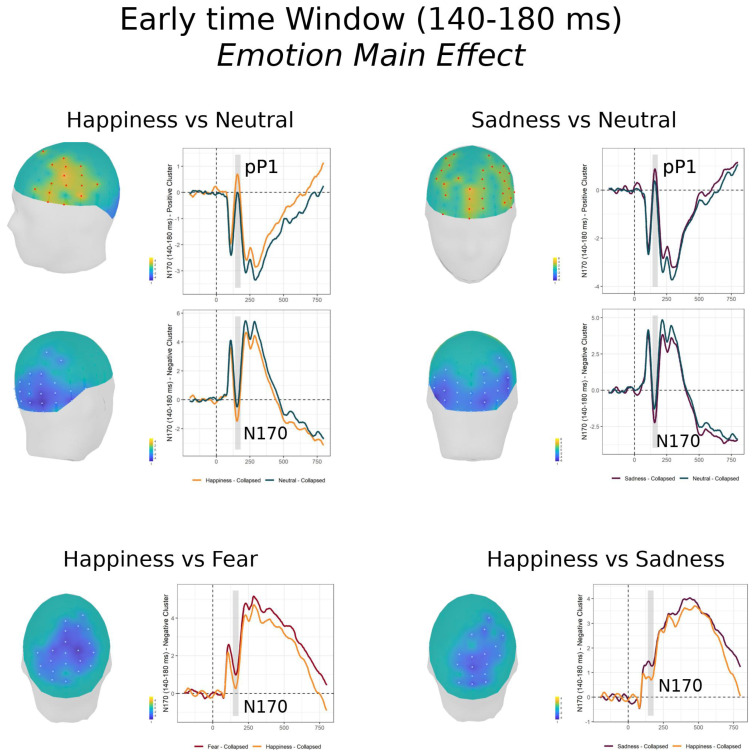
Early time window (140–180 ms): Emotion Main effect: On the right of each panel, grand-average waveforms of the electrodes included in the significant positive and negative clusters; on the left of each panel, the scalp topography of the significant clusters (the sensors included are highlighted in red for the positive clusters and in white for the negative clusters). Left panel: Happiness vs. Neutral; middle panel: Sadness vs. Neutral; right panel: A—Happiness vs. Fear; B—Happiness vs. Sadness. Emotions are collapsed across Conditions.

**Figure 3 brainsci-11-00942-f003:**
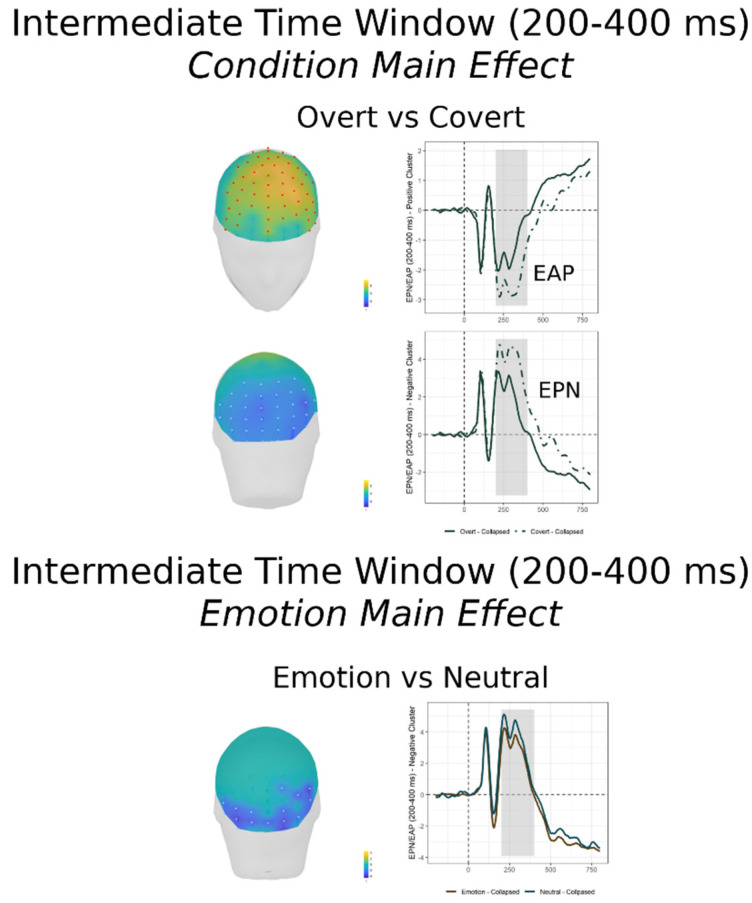
Intermediate time window (200–400 ms): On the right of each panel, grand-average waveforms of the electrodes included in the significant positive and negative clusters; on the left of each panel, the scalp topography of the significant clusters (the sensors included are highlighted in red for the positive clusters and in white for the negative clusters). Left panel: Overt vs. Covert, collapsed across emotions; right panel: Emotion vs. Neutral, collapsed across conditions.

**Figure 4 brainsci-11-00942-f004:**
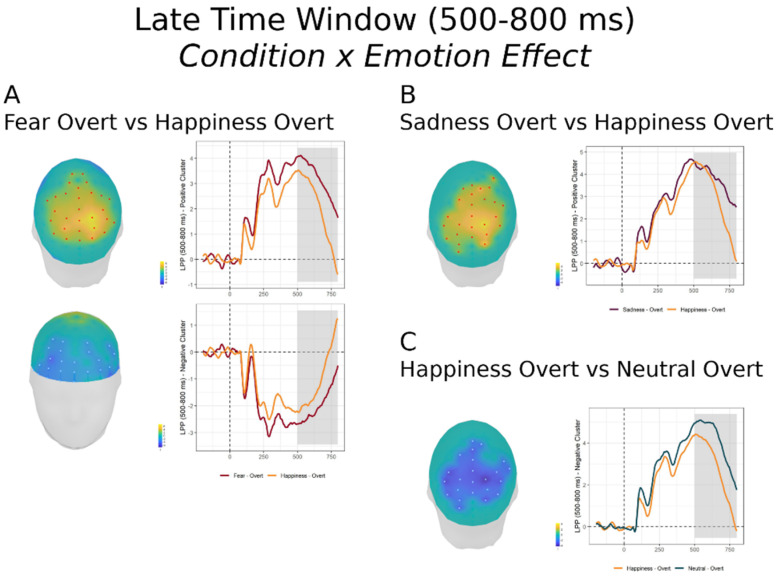
Late time window (500–800 ms): Condition x Emotion effect: On the right of each panel, grand-average waveforms of the electrodes included in the significant positive and negative clusters; on the left of each panel, the scalp topography of the significant clusters (the sensors included are highlighted in red for the positive clusters and in white for the negative clusters). anel (**A**): Fear Overt vs. Happiness Overt; anel (**B**): Sadness Overt vs. Happiness Overt; anel (**C**): Happiness Overt vs. Neutral Overt. Please note that for the latter the direction of the contrast with Happy faces is inverted, explaining the negative cluster.

**Table 1 brainsci-11-00942-t001:** Mean RTs (ms) and standard deviations by Condition and Emotion.

	Happy	Fearful	Sad	Neutral
Covert	715.36 (96.62)	725.70 (110.93)	733.05 (103.55)	724.25 (102.44)
Overt	771.78 ^$^ (93.16)	918.58 ^&^ (107.35)	884.82 ^&^ (96.02)	831.88 (96.75)

^$^ Shorter than neutral, *p* < 0.05; ^&^ longer than neutral, *p* < 0.05.

**Table 2 brainsci-11-00942-t002:** Post-Hoc pairwise mass-univariate tests showing significant negative and positive clusters.

**Early Window (140–180 ms)**	**Negative**	**Cluster**			**Positive**		**Cluster**	
Emotion main effect	Mean A (SD)	Mean B (SD)	*t*-Mass	*p*-Value	Mean A (SD)	Mean B (SD)	*t*-Mass	*p*-Value
Happy vs. Neutral	−1.1 (1.1)	−0.2 (1.0)	1291	0.02	0.4 (0.5)	−0.2 (0.5)	957	0.04
Sad vs. Neutral	−1.99 (1.1)	−1.0 (1.1)	1396	0.002	0.71 (0.5)	0.2 (0.5)	811	0.021
Happy vs. Sad	0.9 (0.8)	1.5 (0.78)	929	0.004	--	--	--	--
Happy vs. Fear	0.6 (0.8)	1.3 (0.8)	1372	0.004	--	--	--	--
**Intermediate Window (200–400 ms)**	**Negative**	**Cluster**			**Positive**		**Cluster**	
Condition main effect	Mean A (SD)	Mean B (SD)	*t*-Mass	*p*-Value	Mean A (SD)	Mean B (SD)	*t*-Mass	*p*-Value
Overt vs. Covert	1.76 (0.8)	3.8 (0.8)	14,094	0.002	−1.1 (0.5)	−2.3 (0.6)	20472	0.002
Emotion main effect								
Emotion vs. Neutral	2.46 (0.9)	3.1 (0.9)	3021	0.03	--	--	--	--
**Late Window (500–800 ms)**	**Negative**	**Cluster**			**Positive**		**Cluster**	
Condition × Emotion	Mean A(SD)	Mean B(SD)	*t*-Mass	*p*-Value	Mean A (SD)	Mean B (SD)	*t*-Mass	*p*-Value
Happy Ov vs. Cov	−4.1 (0.9)	−2.3 (0.9)	8092	0.005	3.5 (0.7)	2.1 (0.8)	7947	0.007
Sad Ov vs. Cov	−4.4 (0.9)	−2.3 (0.9)	10,225	0.004	3.4 (0.8)	1.9 (0.8)	14,471	0.002
Fear Ov vs. Cov	−5 (0.9)	−2.6 (0.9)	8586	0.01	4 (0.7)	2 (0.8)	18,176	0.002
Neutral Ov vs. Cov	−3.8 (0.9)	−1.9 (0.9)	9611	0.009	3.3 (0.8)	1.6 (0.8)	19,889	0.002
Fear Ov vs. Hap Ov	−1.9 (0.8)	−0.9 (0.9)	5362	0.01	3.3 (0.8)	2 (0.8)	8901	0.002
Sad Ov vs. Hap Ov	--	--	--	--	3.9 (0.8)	3 (0.8)	3683	0.023
Hap Ov vs. Neut Ov	4.2 (0.8)	2.8 (0.8)	5809	0.007	--	--	--	--

Top: Early time window (140–180 ms), main effect of Emotion. Center: Intermediate time window (200–400 ms), main effects of Condition (above) and Emotion (below). Bottom: Late time window (500–800 ms), Condition × Emotion interaction. In the columns, for each cluster, Mean amplitude and Standard Deviation (SD) for the first (A) and the second member (B), cluster size, and *p*-value.

## Data Availability

Data are stored and kept in archived form by the supervisor of the study (M.L.).

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
