# Peer review of "Spatiotemporal Dynamics of Covert Versus Overt Processing of Happy, Fearful and Sad Facial Expressions"

_brainsci, 2021, doi:10.3390/brainsci11070942_

Round 1

Reviewer 1 Report

Overall, this is an interesting study. In particular, I like the FMUT analytical approach. I hope that my comments will be helpful for the authors to improve their manuscript.

Introduction

1.It is not easy to understand how the study contributes to the theory. Elaborating on theoretical aspects would benefit a reader about the general idea of the study. If the study aimed to test inconsistencies in the previous empirical research, the correspondence between the previous task and the novel task should be explained and discussed explicitly.

  1. Throughout the manuscript, there is a conceptual confusion between emotion recognition and emotion categorisation. Some consistency is required. 

Methodology and results

  1. EEG studies investigating the extent to which colour information can influence the time course of attentional bias indicate a strong effect of colour on the EPN (e.g., Bekhtereva &Muller, 2017; Yoto et al., 2007; Chai et al., 2019). There is also mounting evidence of the priming effects of colour on emotion categorisation (e.g., Kunieski et al., 2015) including a theory of colours’ impact on psychological functioning (Elliot & Maier, 2012) and studies on emotions and colour preference/avoidance (e.g., Moller et al., 2009; Payen et al., 2011). Therefore, the rationale of using colours and detailed analysis of possible interactions between colours and emotional stimuli is a crucial component for the validity of the task. In addition, it is good practice to check the internal consistency of the measurements (e.g., using the split-half approach). This would support the claim about behavioural results.
  2. The manuscript lacks some critical information about their experimental procedure that limits the reproducibility of their work. For example, were participants’ responses locked to the offset of the stimuli? If so, response speed could not be taken as a reliable measure of performance. If participants were instructed to respond as soon as possible, how the authors handled fast responses (<200 ms)? It is also unclear why the stimuli were presented at a much shorter time (200 ms) than all previous studies (333 ms and greater). The authors could justify the short onsets to convince a reader that participants could process complex information in faces, such as emotional expressions. This is particularly important for interpreting the covert task. 
  3. The claim about the novelty of the task is hardly supported. A similar covert task was used in Wronka & Walentovska (2011). Furthermore, this task is not a modification of an eStroop task as there is no conflict information in the stimuli. Generally, the statements about eStroop inference should be revised as there is no evidence about the inference effect. 
  4. In the introduction, the authors stressed the effects of gender on emotion categorisation. However, their experimental design is imbalanced in terms of the gender of faces. Additional analysis on the effect of gender would provide more robust support for their claims. 
  5. In the results section, no information is provided on effect sizes and confidence intervals. In the present study, with only 16 participants, calculating CIs will provide a reader with important information about the rate at which the CI contains the true population value and power that is closely related to estimation precision. 
  6. Figures 2, 3 and 4 have poor resolution.

Discussion

  1. The discussion could be more focused. For example, the FMUT approach is very interesting. However, the results obtained in this study are very different from comprehensive work on emotion processing using the same analytical approach (Durston & Itier, 2017). 
  2. Some statements are difficult to relate to the results of the study (e.g., the general statement about sensorimotor integration and early visual processing). Overall, the discussion is speculative in places. For example, the effect of valence has never been tested but discussed as a finding. It could be more tightened to the results and the extent of neural differences between conditions. 

11.The author may also wish to give a more detailed discussion of their tasks and highlight advantages and limitations. 

Finally, as a reader, I found the writing style is overcomplicated. In places, the language is unclear, making it difficult to follow. I advise the authors work with a writing coach or copyeditor to improve the flow and readability of the text.

Author Response

Replies to Reviewer 1 

We thank the reviewer per the opening comment that overall the study is interesting, and for the helpful comments to improve the revised manuscript.

Introduction

1a- “It is not easy to understand how the study contributes to the theory… Elaborate more on theoretical aspects….”We have attempted to do so throughout the introduction and the discussion, in the highlighted passages.

1b- “If the aim of the study is to test inconsistencies in previous research, the comparison between the previous task and the novel task should be clarified…. The novel task should be explained and discussed explicitly”. Now we make clearer that the two tasks used in the present study employed the same face expressions contrasting a condition of perceptual distraction (the Covert Task) to one of Attention to Emotion (the Overt Task). We have described in more detail the difference between our Covert task and the Covert task employed by the 3 papers in the literature with a within-subject comparison of task demands (Wronka & Walentovska, 2011; Rellecke et al, 2012, Itier and Tavarez, 2017). These changes are highlighted in the Abstract, Introduction page 3 and 8 and Discussion page 20).

We have also changed Figure 1 to illustrate more clearly the difference between the Covert and the Overt Task.

2- About the “conceptual confusion between emotion recognition and emotion categorization”. Here I believe the reviewer is pointing at the view that perception of emotion expressions is not necessarily categorical, and then it does not necessarily result in the use of discrete emotion labels. In this sense there could be a dissociation between emotion categorization and emotion identification. In our study we test emotion categorization, and we made it more clear in this revision.

Methodology and Results

3- “EEG studies indicate a strong effect of colour on the EPN… There is evidence of priming effects of colour on emotion categorization… Therefore the rationale of using colours and detailed analysis of possible interactions between colours and emotional stimuli is a crucial component for the validity of the task”. I would agree that colour is an important factor to bias early visual attention towards perceptual salience of emotional stimuli, along with other factors (like spatial frequency). This is certainly true for emotional scenes (Bekthereva & Mullet, 2017). I also agree that this is an interesting venue for future research, also given the fact that facial expressions differ from visual scenes in that they are biological stimuli of evolutionary value necessary for social communication. But I would respectfully disagree with the reviewer about the need of exploring the relationship between colour and emotion in this study. Unlike Bekthereva & Mullet study, the present experiment did not include B&W faces, so that the early ERP modulation for colour stimuli could not be evaluated. We used the original Karoliniska set of facial expressions in colour, also used by Rellecke et al (2012) as well as several other studies employing the NITSTIM colour stimulus set (e.g., Smith et al, 2013). Given the concern earlier expressed with the experimental design of the task, I also guess that the reviewer (and therefore also potential other readers) may have not clearly understood the experimental task. We now explain more clearly the rationale and details of the task, and present a modified Figure 1 to show more clearly the task in its Covert and Overt versions. Both conditions involve facial expressions with a small colour square at fixation, on the tip of the nose which are present concomitantly. In the Covert Task, emotions are task irrelevant, while the colour of the square is the task relevant dimension for the task. It could be more properly described as a “Perceptual Distraction” condition, in that subjects are instructed to pay attention to the central color and ignore the surrounding faces. In contrast the Overt condition can be described as an “Attention to Emotion” condition. The behavioural results in the present study are quite similar to those of the three reference papers utilizing the GD Task and the ED Task as Covert and Overt Conditions (Wronka & Walentovska, 2011; Rellecke et al, 2012; Itier and Tavarez, 2017: no RT effects in the Covert, RT differences in the Overt). Therefore, we have no reasons to doubt the RT findings in our study. Lastly, the influence of colour on emotion categories should be studied first behaviorally (if an aim of the study), since ERPs require many repetitions of the same stimuli, and this kind of study would require extracting subaverages for male and female portrayers and therefore acquiring more data to achieve reliable ERP effects. We believe that both these scenarios (behaviour and ERPs) are beyond the scope of the present study.

  1. “The manuscript lacks some critical information about the experimental procedure..”We apologize for missing such information and thank the reviewer for pointing it out. We have added this information in the methods section and highlighted it. First of all, ERPs were time-locked to stimulus onset. Furthermore, RTs were measured from stimulus onset and not offset. We included in the analysis only RT between 200 and 2000 msec. Eliminating responses shorter than 200 ms was chosen nor only to rule out guesses or premature responses but also to control for eye movements, given the known latency of saccadic movements. This is also the reason to present the stimuli for “only” 200 msec, to discourage participants to perform eye movements which generate huge electrical artefacts. The reviewer states that “all previous studies had longer presentations of more than 333 ms”. I don’t share this opinion: in fact Itier and Tavares, 2017, similarly concerned with eye movements, showed their faces for 260 ms, and Wronka and Walentowska, 2011 for 300 msec. The reviewer states that “short onsets may affectparticularly the covert task”. Well, many ERP studies of covert emotional processing in faces exploring attentional bias for threat information have indeed used very short presentations as a proof that fear and threat can be processed early and preattentively (some of them even subliminally using below recognition threshold followed by masks). Many ERP studies of visual attention use visual search arrays presented for 100 ms. Lastly, for both tasks in the present study we had a practice block that ended when subjects achieved 80% accuracy in both tasks. In the RT results it is reported that Accuracy was 84% or higher in both tasks, and there was no main effect of Task on Accuracy; such Accuracy is comparable to that reported in other studies using longer presentations. This is important also considering that Fear and Sad expressions were somewhat harder to discriminate.

5a. “The claim about novelty of the task is hardly supported. A similar covert task was used in Wronka & Walentovska, 2011)” When we state our main hypotheses we report three studies in the literature that had explored the effect of top-down attention in emotion processing using two tasks (Wronka & Walentovska, 2011; Rellecke et al, 2012; Itier and Tavares, 2017). All of them employ as Covert Task the Gender Discrimination Task. In the paper we try to build a case that since Gender Discrimination (GD) and Emotion Categorization (ED) are not completely independent, and people are still paying attention to Faces, perhaps a better comparison task (I mean Covert) is a “Perceptual Distraction Task” where people are instructed to perform a different perceptual task (color discrimination of the central square) and to ignore the surrounding faces, to contrast to the Overt task of “Attention to Emotion”, i.e. the Emotion Categorization task. The claim about novelty concerns the use of the Covert Task in the present study rather than the GD task in the previous above mentioned studies: for us a better match to test the role of Attention in Emotional processing.

5b. “Furthermore, this task is not a modification of a eStroop Task as there is no conflict information in the stimuli. Generally, the statements about eStroop inference(?) should be revised  as there is no evidence about the inference(?)effect”. We are sorry for the confusion generated in this reviewer. Although we still believe that the covert task is a modified emotional Stroop task, since this aspect is not critical and may generate similar confusion in the reader, we have removed any reference to the eStroop task and its interpretation as such in the revised paper. We have rather described the covert task as a perceptual distraction task where emotional faces are task-irrelevant, in contrast with the Attention to Emotion task where emotions are task relevant and emotion categorization is required.

  1. In the introduction the authors stressed the effects of gender on emotion categorization. However, their experimental design is unbalanced in terms of the gender of the faces. Additional analysis on the effect of gender would provide more robust support of their claims”.We mentioned the implicit effect of gender to justify the fact that the Covert Gender Discrimination Task employed in the previous studies may not be the best task to employ as comparison to the Overt Gender Discrimination Task (not entirely independent). We now recognize as a limitation of the present study the fact that the emotional faces were slightly unbalanced in terms of gender representation (13 males, 15 females); we did start off with an equal number of male and female portrayers but found that in this dataset females were slightly better at conveying emotions and preferred to have only the most agreed upon expressions. We did not employ different event codes to male and female expressions, therefore we cannot create now selective ERP averages as a function of gender of the encoder. In addition, the effect of gender was not an aim of the study, exploring the time course of the effect of top-down attention to Emotion on Emotion processing. We also propose that gender of the portrayer should be included in a future larger size study where the addition of a new variable (gender of the encoder) would be compensated by a larger number of trials to allow reliable ERP effects, Limitations and future directions, page 30.
  2. No information is provided on effect sizes and confidence intervals, important in a study with only 16 participants”. We thank the reviewer for this useful comment. We have now added effect sizes for main F-MUT effects in the main text (partial η2) and we have included a new supplementary Table 3 reporting Coehen’s d and Confidence Intervals for the significant Post-Hoc contrasts discussed in the paper.
  3. “Figure 2, 3 and 4 have poor resolution”. We have generated and included new high resolution tiff format images for these figures.

Discussion

  1. “The discussion should be more focused. The FMUT approach is very interesting… However the results are vey different from another study using the same approach (Durston & Itier, 2017)”.We thank the reviewer for the positive comment on the FMUT approach and have tried to focus more the discussion. The conclusion about the discrepancy of the findings of Durston and Itier (2021) it’s not granted though, because they had similar results to ours in terms of Task main effect and Emotion main effect and lack of a Significant Task X Condition interaction in the time window examined (0-356 ms). Their N170 and EPN effects are merged together due to their choice of a continuous time-window, while we subdivided the time window into three sub-epochs. We have included a comment on Itier’s F-MUT re-analysis of the previous data from Itier and Tavares (2017) in the conclusion (page 30).

10a. “Some statements are difficult to relate to the results of the study (e.g. the general statement about sensorimotor integration and early visual processing…..”We have removed the sentence about sensorimotor integration, just left the reference to the pP1 as frontal concomitant of the visual N1.

10b. “Overall the discussion is speculative in places. For example, the effect of valence has never been tested but discussed as a finding. It could be more tightened to the results and the extent of the neural differences between conditions”.We have tried to remove more tangential comments and limited to try to interpret the significant results. We have removed the references to valence effects. In a couple of instances though, to stress the theoretical implications, we have reported our interpretations qualifying them by: ‘we can speculate…): Discussion, pages 23 and 24.

11a. “The authors may wish to give a more detailed discussion of their tasks and highlight advantages and limitations”.We have tried to do so throughout the paper and also in the conclusions. We have modified the Figure 1 to illustrate better the tasks.

11b. “As a reader, I found the writing style overcomplicated…. Making it difficult to follow… I advise to work with a copyeditor to improve the readability of the text”. Well taken, we have tried to simplify the writing style, and have used the services of a scientific copyeditor.

Reviewer 2 Report

 The study raised the question of whether attention to face-unrelated perceptual information or attention to the emotional expression modulates activations towards faces concerning the N170, EPN, or LPP component (see details in the attached PDF).

Author Response

Replies to Reviewer 2

We are thankful to this reviewer for the general assessment that the study addresses an important question and that the methodology used is sound and seems to be carefully performed. 

-“although ~30% trials were rejected”. The reason for this apparently higher rate of rejection is for sake of being more careful and conservative and not because of noisy data or other problems with data collection. In particular, we opted for removing EEG epochs with blink artefacts and eye movements rather than applying an ICA approach to deconvolve eye movement activity and regenerate the EEG over frontal channels. We did it for two reasons:

  1. a) in trials contaminated by eye blinks or saccadic activity it has been shown that activity in visual cortex is inhibited; obviously people are not attending or possibly not even perceiving the stimulus;
  2. b) we were interested in frontal effects accompanying ERP emotion modulations, and were concerned with the fact that the ICA for eye-artefact removal as other multivariate methods could possibly unduly remove brain activity over frontal regions typically affected by eye movement artefacts.

 “Many references are made towards pictures or verbal material in the introduction (and sensorimotor integration in the discussion) and fMRI studies…..  It would be highly beneficial to depict findings from the extraordinarily rich field of ERP face studies” We agree with the reviewer and in the revised paper focused our attention just on ERP studies of emotional faces.

 “The lack of N170 increases for fearful faces is unexpected. I would discuss that this can be due to the small sample, specific fearful images, or for the presentation in the same task of fearful and sad faces, hard to discriminate from each other….It reads as if an N170 effect is equally often found or not found, while the literature shows the strongest/most frequently reported effect for fearful faces”. Once again we totally agree with the reviewer and have incorporated his comments and references in the introduction and discussion, including the papers showing N170 fear modulations in conditions of highly demanding perceptual load. 

“In our view the N170 increase is…. an actual emotional component.. .signalling an index of acquired emotional information….”We totally agree with the reviewer interpretation and mention it along with our discussion of category effects in terms of perceived emotional arousal or personal salience (Discussion, page 22 and 25).  

“To balance the discussion, there is a couple of other studies reporting Happy vs, Neutral increases….. Even not affected by a variety of perceptual load levels”.Once again we thank the reviewer and added these studies in the introduction (page 4) and discussion (page 23).

“The tasks compared in the study are highly interesting, contrasting perceptual distraction vs. attention to emotion…. I would also say that some more studies compare at least two tasks, or even more than two task conditions”We thank the reviewer for her/his appreciation of the tasks chosen.  We have now included the additional references to studies employing two or more tasks (introduction page 5, 7)

“I think it is incorrect to state that during the EPN there was greater activity when emotional expressions were overtly attended given that there were only separate main effects of task and emotion..”We have now corrected our statements in the abstract and discussion.

“Currently it reads as the findings of the mass univariate approach would serve as ground truth…. It has advantages and disadvantages.. I would lower the tone a bit”Absolutely well taken. We qualify in the general conclusions the trade-offs of the mass univariate approach in the Limitations and future directions, page 31.

“I would remove the Helsinki statement…. Including only local ethics approval”. Done, we weren’t aware of this new policy and thank the reviewer.

“The figures should be enhanced in resolution and/or size”. We include now high-resolution tiff format versions of Figure 2,3,4.

“Readers may be interested in how each of the emotion waveforms looks during the EPN… A better visualization would be for each emotion effect to provide waveforms and topo differences for all conditions next to each other… showing all data”. We now add supplementary Figure 5 and 6 that show overlaid grandaverage waveforms for each emotion for the covert and overt condition for the N170 and EPN at occipital and frontal sites, and for the LPP at parietal and occipital sites. Difference waves are omitted because they would involve a combination of 4 emotions x 2 conditions….

“The epoch should not end directly at the end of the examined interval but rather around 1000 msec”.We agree that it would have been better. Unfortunately, while the EEG data were acquired continuously, we segmented epochs starting from -200 msec to 800 msec, therefore could not have displayed ERPs past 800 msec.

 “Taken together I think the study uses an exciting paradigm and asks a relevant question…”. We are very thankful to the reviewer for this flattering comment, and thank the reviewer for the long list of references added for our benefit.

Round 2

Reviewer 1 Report

all my comments were addressed

Reviewer 2 Report

Thank you very much for the great revision. I found the study already really interesting, and I think it further improved. The Supplementary Figures help further to have a look at the ERP waveforms apart from statistics.

Just as a comment, and it is up to the authors if they even want to discuss it. I noted that fearful faces had strong left/right N170 differences, i.e., the largest N170 amplitudes over the right but smallest over the left sensors. They also had the largest positivity during the P1 in the overt condition (Supplementary Figure 2), and those arguing for peak-to-peak measures for early ERPs might say that the subsequent N170 is underestimated. I am not fully subscribing to this view. Still, when discussing the variability or heterogeneity of (e.g. N170) findings, such earlier ERP differences and/or differences in lateralizations could be further influencing factors (when they seem to differ across emotional expressions and might even be task-dependent). 

Finally, while I cannot provide detailed language corrections, I spotted an error on page 27 it likely should read "acquired" instead of "accrued"

Congratulations on this interesting paper, looking forward to future research and with best wishes, Sebastian Schindler